# Cost-effectiveness of avelumab first-line maintenance therapy for adult patients with locally advanced or metastatic urothelial carcinoma in France

Fanny Porte[1], Anna Granghaud[2¤a], Jane Chang[3], Mairead Kearney[4], Aya Morel[2], Ingrid Plessala[5]*, Hélène Cawston[5], Julie Roiz[6], Ying Xiao[6], Marie-Noelle Solbes[7], Prisca Lambert[8], Alain Ravaud[9], Yohann Loriot[10], Antoine Thiery-Vuillemin[11¤b], Pierre Lévy[12]

1 Health Economics Department, Merck Santé S.A.S., Lyon, France, 2 Health Economics department, Pfizer S.A.S., Paris, France, 3 Health Economics department, Pfizer, New York, NY, United States of America, 4 Global Value Demonstration, Market Access and Pricing, The Healthcare Business of Merck KGaA, Darmstadt, Germany, 5 Health Economics & Market Access (HEMA), Amaris Consulting, Paris, France, 6 Health Economics, Evidera, London, United Kingdom, 7 Medical Department, Merck Santé S.A.S., Lyon, France, 8 Medical Department, Pfizer S.A.S., Paris, France, 9 Department of Medical Oncology, Centre Hospitalo-Universitaire de Bordeaux, Bordeaux, France, 10 Department of Medical Oncology, Institute Gustave-Roussy, Villejuif, France, 11 Department of Medical Oncology, Centre Hospitalier Régional Universitaire de Besançon, Besançon, France, 12 Université Paris-Dauphine, Université-PSL, [LEDA], LEGOS, Paris, France

¤a Current address: AbbVie, Rungis, France
¤b Current address: Bristol Myers Squibb, Boudry, Switzerland
* ingrid.plessala@amaris.com

**Data Availability Statement:** Any requests for data by qualified scientific and medical researchers for legitimate research purposes will be subject to the

## Abstract

### Background

This study evaluated the cost-effectiveness of avelumab first-line (1L) maintenance therapy plus best supportive care (BSC) versus BSC alone for adults with locally advanced or metastatic urothelial carcinoma (la/mUC) that had not progressed following platinum-based chemotherapy in France.

### Methods

A three-state partitioned survival model was developed to assess the lifetime costs and effects of avelumab plus BSC versus BSC alone. Data from the phase 3 JAVELIN Bladder 100 trial (NCT02603432) were used to inform estimates of clinical and utility values considering a 10-year time horizon and a weekly cycle length. Cost data were estimated from a collective perspective and included treatment acquisition, administration, follow-up, adverse event–related hospitalization, transport, post-progression, and end-of-life costs. Health outcomes were measured in quality-adjusted life-years (QALYs) and life-years gained. Costs and clinical outcomes were discounted at 2.5% per annum. Incremental cost-effectiveness ratios (ICERs) were used to compare cost-effectiveness and willingness to pay in France. Uncertainty was assessed using a range of sensitivity analyses.

healthcare business of Merck KGaA, Darmstadt, Germany's (CrossRef Funder ID: 10.13039/ 100009945) Data Sharing Policy. All requests should be submitted in writing to the healthcare business of Merck KGaA, Darmstadt, Germany's data sharing portal (https://www.emdgroup.com/ en/research/our-approach-to-research-and-development/healthcare/clinical-trials/ commitment-responsible-data-sharing.html). When the healthcare business of Merck KGaA, Darmstadt, Germany has a co-research, co-development, or co-marketing or co-promotion agreement, or when the product has been out-licensed, the responsibility for disclosure might be dependent on the agreement between parties. Under these circumstances, the healthcare business of Merck KGaA, Darmstadt, Germany will endeavor to gain agreement to share data in response to requests.

**Funding:** This study was funded by Merck Santé S. A.S, an affiliate of Merck KGaA, Darmstadt, Germany (CrossRef Funder ID: 10.13039/ 100009945), as part of an alliance between the healthcare business of Merck KGaA, Darmstadt, Germany and Pfizer. Authors who were employees of the funders were involved in the model design, data collection, analysis, manuscript preparation, and the decision to publish.

**Competing interests:** F. Porte is an employee of Merck Santé S.A.S., Lyon, France, an affiliate of Merck KGaA, Darmstadt, Germany at the time of the project. A. Granghaud was an employee of Pfizer S.A.S., Paris, France at the time of the study. J. Chang is an employee of Pfizer and holds stock and other ownership interest with Bayer, Bristol Myers Squibb, and Pfizer. M. Kearney is an employee of Merck KGaA, Darmstadt, Germany, and holds stock in Merck KGaA, Darmstadt, Germany, Novartis and UCB. A. Morel is an employee of Pfizer S.A.S., Paris, France. I. Plessala was an employee of Amaris Consulting, Paris, France at the time of the study. H. Cawston is an employee of Amaris Consulting, Paris, France. J. Roiz is an employee of and reposts stocks and other ownership interest with Evidera. Y. Xiao is an employee of Evidera. M.-N. Solbes is an employee of Merck Santé S.A.S., Lyon, France, an affiliate of Merck KGaA, Darmstadt, Germany. P. Lambert is an employee of Pfizer S.A.S., Paris, France. A. Ravaud has received grants or contracts from Merck KGaA, Darmstadt, Germany, and Pfizer; has received travel and accommodation expenses from Ipsen Merck KGaA, Darmstadt, Germany, Merck & Co., Kenilworth, NJ, and Pfizer; and has participated in advisory boards for Esai, Ipsen, Merck KGaA, Darmstadt, Germany, and Pfizer. Y.

## Results

Avelumab plus BSC was associated with a gain of 2.49 QALYs and total discounted costs of €136,917; BSC alone was associated with 1.82 QALYs and €39,751. Although avelumab plus BSC was associated with increased acquisition costs compared with BSC alone, off-sets of −€20,424 and −€351 were observed for post-progression and end-of-life costs, respectively. The base case analysis ICER was €145,626/QALY. Sensitivity analyses were consistent with the reference case and showed that efficacy parameters (overall survival, time to treatment discontinuation), post-progression time on immunotherapy, and post-pro-gression costs had the largest impact on the ICER.

## Conclusions

This analysis demonstrated that avelumab plus BSC is associated with a favorable cost-effectiveness profile for patients with la/mUC who are eligible for 1L maintenance therapy in France.

## Introduction

Urothelial carcinoma (UC), also known as transitional cell carcinoma, is the most common type of bladder cancer worldwide and the seventh most common cancer in France, accounting for approximately 90% of all bladder cancer cases [1–3]. UC is the most frequent morphological form of cancer of the excretory system and can cause malignancies in the renal calyces, renal pelvis, ureter, bladder, and urethra [2].

Bladder cancer is more common in men than women and is the fourth most prevalent cancer in men in France [4]. The median age at diagnosis is 73 years in men and 78 years in women [5]. According to the latest report by the French National Cancer Institute, published in 2019, 13,074 incident cases of bladder cancer were reported in France in 2018, of which 81% were in men [5]. Owing to nonspecific symptoms, diagnosis is often delayed. Symptoms of early-stage UC may include hematuria (initially detected by unexplained anemia) and signs of irritation, such as dysuria, burning during urination, or frequent urination; symptoms of advanced bladder cancer may include pyuria and pelvic pain [6]. Locally advanced or meta-static UC (la/mUC) can negatively affect patients' physical functioning, leading to an impaired quality of life (QoL) and reduced life expectancy; thus, treatment aims to improve overall sur-vival (OS) and QoL.

Avelumab is a human immunoglobulin G1 anti–programmed death-ligand 1 (PD-L1) monoclonal antibody that was approved by the European Medicines Agency on January 21, 2021, as monotherapy for the first-line (1L) maintenance therapy of adults with la/mUC who are progression free after receiving platinum-based chemotherapy. Before the availability of avelumab, no maintenance therapy was recommended for patients with la/mUC, and patients with stable disease or objective response after completing platinum-based 1L chemotherapy received supportive care and were managed via a 'watch-and-wait approach' until disease pro-gression [7]. The approval of avelumab as 1L maintenance therapy for la/mUC was based on the results of the JAVELIN Bladder 100 study (NCT02603432) [8], a randomized, open-label, multicentric, phase 3 trial comparing the efficacy of avelumab 1L maintenance plus best sup-portive care (BSC) versus BSC alone in adults with la/mUC that had not progressed with plati-num-based chemotherapy. The study enrolled 350 patients in each treatment arm and the

Loriot has served in consulting or advisory roles for Astellas Pharma, Bristol Myers Squibb, Immunomedics, Janssen, Loxo/Lilly, Merck KGaA, Darmstadt, Germany, Merck & Co., Kenilworth, NJ, Pfizer, Roche, Seattle Genetics, and Taiho Pharmaceutical; has received travel and accommodations expenses from Astellas Pharma, Janssen Oncology, Merck & Co., Kenilworth, NJ, Roche, and Seattle Genetics; and has received institutional research funding from Astellas Pharma, Basilea, Bristol Myers Squibb, Exelixis, Gilead Sciences, Incyte, Janssen Oncology, Merck KGaA, Darmstadt, Germany, Merck & Co., Kenilworth, NJ, Nektar, Pfizer, Roche, Sanofi, Seattle Genetics, and Taiho Pharmaceutical. A. Thiery-Vuillemin has participated in advisory boards for Astellas Pharma, AstraZeneca, Bristol-Myers Squibb, Ipsen, Janssen, Merck & Co., Kenilworth, NJ, Novartis, Pfizer, Roche/Genentech and Sanofi; reports employment by Bristol Myers Squibb; has served on steering committees for AstraZeneca, Bristol-Myers Squibb and Novartis; has received institutional research funding from Bayer, Ipsen and Pfizer; has served as principal investigator for Astellas Pharma, AstraZeneca, Bristol-Myers Squibb, Excelixis, Incyte, Ipsen, Johnson & Johnson, Merck & Co., Kenilworth, NJ, Novartis, Pfizer, Roche, Sanofi, and UNICANCER/ GETUG; has received travel and accommodation expenses from Astellas Pharma, AstraZeneca, Bristol-Myers Squibb, Ipsen, Johnson & Johnson, Merck & Co., Kenilworth, NJ, Pfizer and Roche; and is a member of ASCO and GETUG. P. Lévy has served in consulting or advisory role and had received honoraria from Merck KGaA, Darmstadt, Germany. This does not alter our adherence to PLOS ONE policies on sharing data and materials

primary endpoint was OS. Study treatment started 4–10 weeks after completing 4–6 cycles of 1L platinum-based chemotherapy with cisplatin or carboplatin plus gemcitabine. The JAVELIN Bladder 100 trial showed that avelumab 1L maintenance plus BSC improved OS compared with BSC alone in patients who did not have disease progression after four to six cycles of cisplatin or carboplatin plus gemcitabine. An OS improvement of 7.1 months was observed (hazard ratio, 0.69; adjusted 95% CI [0.536–0.923]; p = 0.0005, below the predefined threshold of 0.0053) [7–9]. Based on clinical data from the JAVELIN Bladder 100 trial, avelumab 1L maintenance therapy after platinum-based chemotherapy was recommended as the standard of care for eligible patients with la/mUC in the European Society for Medical Oncology Clinical Practice Guidelines from 2020 onwards (level IA evidence, Fig 1) [3]. The guidelines were updated in March 2022 to recommend avelumab 1L maintenance as the standard of care for patients with la/mUC who are cisplatin eligible and have received 1L cisplatin-based chemotherapy or for cisplatin-ineligible patients who have received 1L carboplatin plus gemcitabine, and whose disease did not progress after completing chemotherapy [3]. Avelumab 1L maintenance is also recommended in guidelines from the National Comprehensive Cancer Network, the European Association of Urology, and the Cancer Committee of the French Association of Urology [2, 4, 10–12].

Assessing the economic value of a new drug indication is a mandatory step of the French reimbursement process, whereby the Economic and Public Health Committee (Commission d'Évaluation Économique et de Santé Publique) issues appraisals to support public authorities in pricing negotiations. The conditions of eligibility for an economic evaluation include the claim of an added clinical value (Amélioration du Service Médical Rendu) of level I to III and a significant impact on health insurance expenditure, considering its impact on the organization of care, professional practices, or methods of patient management, and where appropriate, its turnover [13]. The aim of this study was to assess the cost-effectiveness of avelumab 1L maintenance therapy plus BSC compared with BSC alone in adults with la/mUC that had not progressed following platinum-based chemotherapy in France, based on the outcome of the pivotal JAVELIN Bladder 100 study [8]. This assessment supported an application to the French National Authority for Health (HAS), submitted by Merck Santé S.A.S., Lyon, France, an affiliate of Merck KGaA, Darmstadt, Germany, and Pfizer S.A.S, Paris, France before the adoption of avelumab 1L maintenance into treatment guidelines, for inclusion in the hospital formulary list of reimbursed proprietary medicinal products approved for use in this indication at the authorized dose of 800mg every two weeks.

## Materials and methods

### Patients and study design of the JAVELIN Bladder 100 trial

This economic analysis was based on the JAVELIN Bladder 100 study (data cutoff: October 21, 2019) [9]. The study was conducted at 197 sites (99 in Europe, including 17 in France) and enrolled 700 patients with histologically confirmed la/mUC who had completed 4 to 6 cycles of platinum-based chemotherapy followed by an interval of 4 to 10 weeks. Patients were randomized (1:1) to receive either avelumab 1L maintenance (10 mg/kg intravenously every 2 weeks) plus BSC or BSC alone. A total of 82 French patients were included in the study, making France the second largest patient population by country in the trial. The characteristics of patients in JAVELIN Bladder 100 were compared with data from a retrospective observational study (chart review) conducted as part of the present analyses to ensure that they were comparable with the French population. The retrospective study used medical records from 206 French patients aged ≥18 years, with advanced UC who had received 1L treatment with gemcitabine and a platinum agent without disease progression (Table 1) [14]. Based on these data,

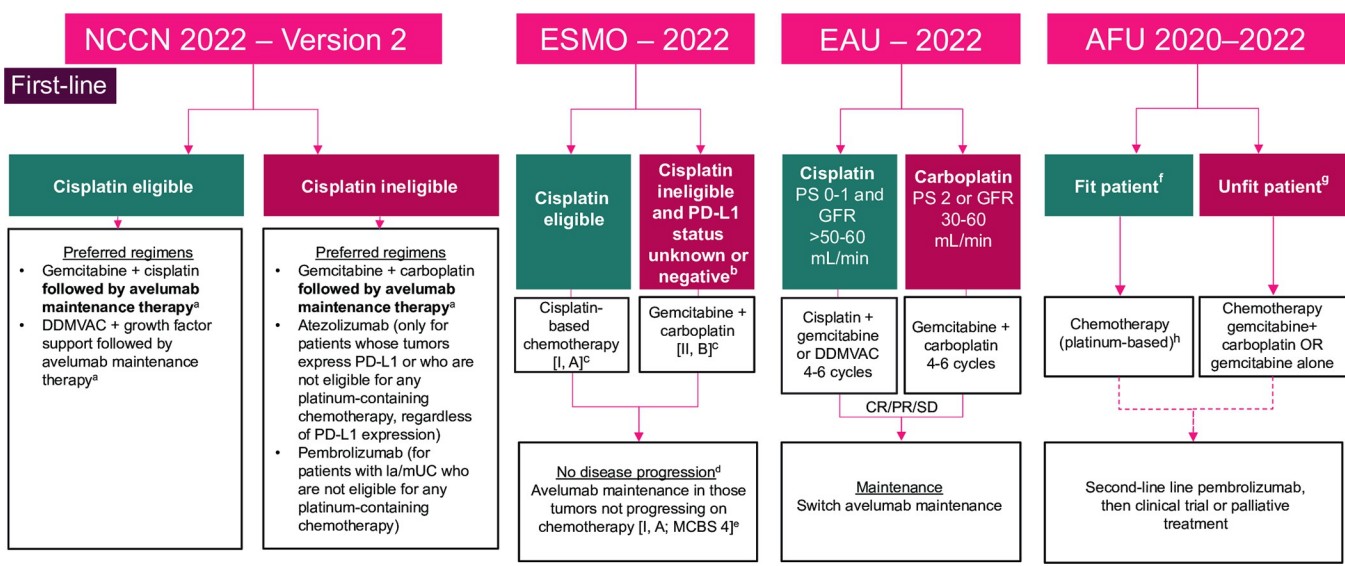

**Fig 1. Current recommendations for the 1L treatment of la/mUC.** 1L, first-line; AFU, French Urological Association; CR, complete response; DDMVAC: dose-dense methotrexate, vinblastine, doxorubicin, and cisplatin; EAU, European Association of Urology; EMA, European Medicines Agency; ESMO, European Society for Medical Oncology; FDA, US Food and Drug Administration; GFR, glomerular filtration rate; la/mUC, locally advanced or metastatic urothelial carcinoma; MCBS, Magnitude of Clinical Benefit Scale; NCCN, National Comprehensive Cancer Network; PD-L1, programmed death-ligand 1; PS, performance status; PR, partial response; SD, stable disease. [a] Maintenance therapy with avelumab only if there is no progression with 1L platinum-containing chemotherapy. [b]Creatinine clearance grade 2 and New York Heart Association class III heart failure. [c] Rechallenge with platinum-based chemotherapy may be considered if progression occurred 12 months after the end of previous platinum-based chemotherapy or 12 months after the end of previous platinum-based chemotherapy and avelumab maintenance therapy. [d] This should be assessed within 10 weeks of completion of chemotherapy. [e] ESMO-MCBS v1.1120 was used to calculate scores for new therapies and indications approved by the EMA or the FDA. The scores have been calculated by the ESMO-MCBS Working Group and validated by the ESMO Guidelines Committee (https://www.esmo.org/guidelines/esmo-mcbs/scale-evaluation-forms-v1.0-v1.1/scale-evaluationforms-v1.1). [f] Fit patient: creatinine clearance ≥60 mL/min and PS <2. [g] Unfit patient: creatinine clearance <60 mL/min and PS≥2. [h] Patients experience no progression after 1L chemotherapy (stable or responsive disease). Maintenance avelumab has shown a benefit in overall survival (pending availability of the molecule).

the comparability of JAVELIN Bladder 100 patient characteristics to French patients with la/mUC was confirmed (Table 1). The primary objective of the JAVELIN Bladder 100 study was to demonstrate the superiority of avelumab 1L maintenance plus BSC compared with BSC alone in terms of OS. The median duration of treatment was 24.9 weeks in the avelumab plus BSC arm and 13.1 weeks in the BSC alone arm. Median OS was 21.4 months (95% CI [18.9–26.1]) in the avelumab plus BSC arm versus 14.3 months (95% CI [12.9–17.9]) in the BSC alone arm. Adverse events (AEs) were graded according to the National Cancer Institute Common Terminology Criteria for AEs (CTCAE) version 4.03 [7]. Costs and effects of avelumab plus BSC were compared with those of BSC alone in the absence of other therapeutic alternatives for 1L maintenance of patients with la/mUC according to French and European guidelines [4, 7, 15, 16].

## Model structure

An area under the curve or partitioned survival model was adapted in accordance with French methodological guidelines for the economic analysis [17]. This model included three health states: progression-free survival, post progression, and death (Figs 2 and 3) and was programmed using Excel (Microsoft Corporation) and Visual Basics for Applications. The model also distinguished between patients who were or were not receiving treatment. This distinction only affected the cost calculations and not the efficacy estimation. A weekly cycle length, with a frequency of administration which varied from 2 to 4 weeks, was considered due to the

**Table 1. Patient characteristics from the JAVELIN Bladder 100 study and the retrospective chart review study.**

| | JAVELIN Bladder 100 Trial | | Retrospective Study |
|---|---|---|---|
| | Avelumab plus BSC n = 350 | BSC alone n = 350 | N = 206 |
| **Age, years** | | | |
| Median | 68 | 69 | 66.3 |
| Mean (SD) | 67 (9.5) | 68 (9.2) | 66 (7.3) |
| **Sex, n (%)** | | | |
| Male | 266 (76.0) | 275 (78.6) | 158 (76.7) |
| Female | 84 (24.0) | 75 (21.4) | 48 (23.3) |
| **ECOG status, n (%)[a]** | | | |
| 0 | 213 (60.9) | 211 (60.3) | 31/205 (15.1) |
| 1 | 136 (38.9) | 136 (38.9) | 127/205 (62.0) |
| ≥2 | 1 (0.3%) | 3 (0.9) | 47/205 (22.9) |
| Unknown | 0 | 0 | 0 |
| **Site of the primary tumor, n (%)** | | | |
| Prostate | 3 (0.9) | 0 | 0 |
| Bladder | 236 (67.4) | 264 (75.4) | 167 (81.1) |
| Ureter | 47 (13.4) | 36 (10.3) | 17 (8.3) |
| Urethra | 5 (1.4) | 5 (1.4) | 3 (1.5) |
| Renal pelvis | 59 (16.9) | 45 (12.9) | 19 (9.2) |
| Not reported | 0 | 0 | 0 |
| Other | 0 | 0 | 0 |
| **Metastatic sites, n (%)** | | | |
| Visceral | 191 (54.6) | 191 (54.6) | 134 (65.1)[c] |
| Nonvisceral | 159 (45.4) | 159 (45.4) | 72 (34.9)[c] |
| **PD-L1 status, n (%)** | | | |
| Positive | 189 (54.0) | 169 (48.3) | 18/26 (69.2) |
| Negative | 139 (39.7) | 132 (37.7) | 8/26 (30.8) |
| Unknown | 22 (6.3) | 49 (14.0) | 180/206 (87.4) |
| **First-line chemotherapy regimen, n (%)** | | | |
| Cisplatin + gemcitabine | 183 (52.3) | 206 (58.9) | 111 (53.9) |
| Carboplatin + gemcitabine | 147 (42) | 122 (34.9) | 95 (46.1) |
| Carboplatin + cisplatin + gemcitabine[b] | 20 (5.7) | 20 (5.7) | 0 |
| Not reported | 0 | 2 (0.6) | 0 |

BSC, best supportive care; ECOG, Eastern Cooperative Oncology Group; NA, not applicable; NR, not reported; PD-L1, programmed death-ligand 1; SD, standard deviation.

[a] For the medical chart review, the percentage is calculated among patients for whom ECOG PS was recorded (n = 205)

[b] This includes patients who switched platinum-based regimens while receiving first-line chemotherapy

[c] In the medical chart review, visceral metastases at baseline included distant metastases in the lung, liver, kidney, brain, peritoneum, bladder, pleura, and adrenal glands, based on the data collection.

heterogeneous dosing regimen of therapeutic strategies. A time horizon of 10 years was considered in the base case analysis due to the average age of patients in the study (67.5 years), the aggressiveness of the disease, the estimated proportion of patients alive at 10 years (6% versus 10–18% at 5 years), based on extrapolations of clinical data recommended by the National Institute for Health and Care Excellence Decision Support Unit [18] and the opinions of the clinical experts. A collective perspective was adopted in accordance with French methodological guidelines for the economic analysis. The proportion of men considered in the model and

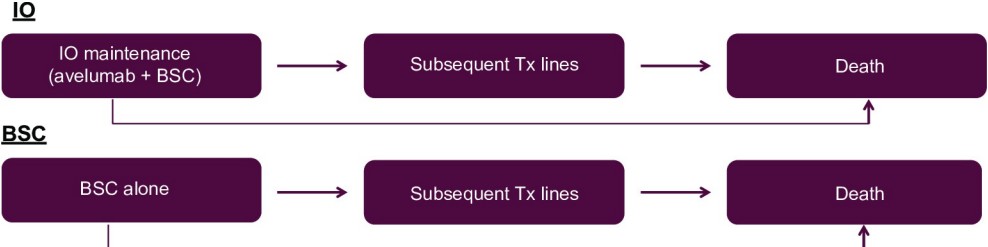

**Fig 2. Schematic diagram of model utilized in the cost-effectiveness analysis.** BSC, best supportive care; IO, immuno-oncotherapy; Tx, treatment.

the mean patient age were aligned with those of the intention-to-treat population in the JAVE-LIN Bladder 100 study.

OS, progression-free survival (PFS), and time-to-treatment discontinuation (TTD) data from the trial were extrapolated beyond the duration of the clinical trial, the trial follow-up (median, 19.6 months) was shorter than the model time horizon (10 years in the base case analysis). Raw OS, PFS, and TTD data from Kaplan-Meier (KM) curves [9] were considered for durations of follow-up in the trial without further parametric modelling (36.34, 32.89, and 35.65 months, respectively), and extrapolations were considered thereafter.

OS data for both arms were extrapolated based on a dependent parametric survival model; the significance value for the test of proportional hazards was >0.05 (p = 0.198), indicating that the proportional hazards assumption was not violated. A log-normal parametric distribution was considered based on Akaike information criteria and Bayesian information criteria, visual inspection of curves, and clinical plausibility. To ensure that the mortality rate of patients with UC was not higher than that of the general French population, the OS of patients with la/mUC in the model was capped using French lifetables [19]. Thus, the modeled hazard of death could not be higher than that of the general population.

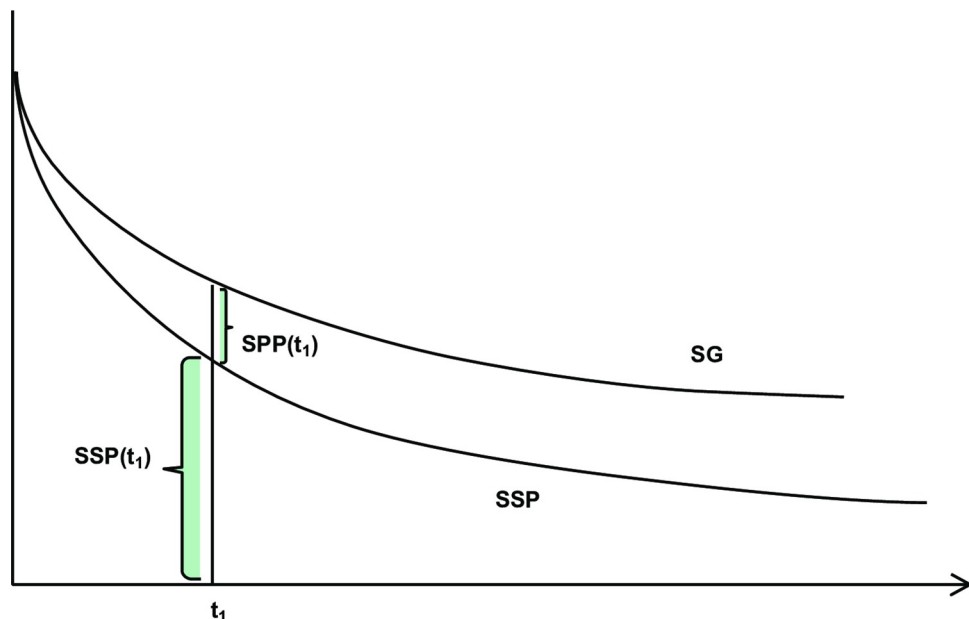

**Fig 3. Area under the curve model developed for the cost-effectiveness analysis.** OS, overall survival; PPS, post-progression survival; PFS, progression-free survival.

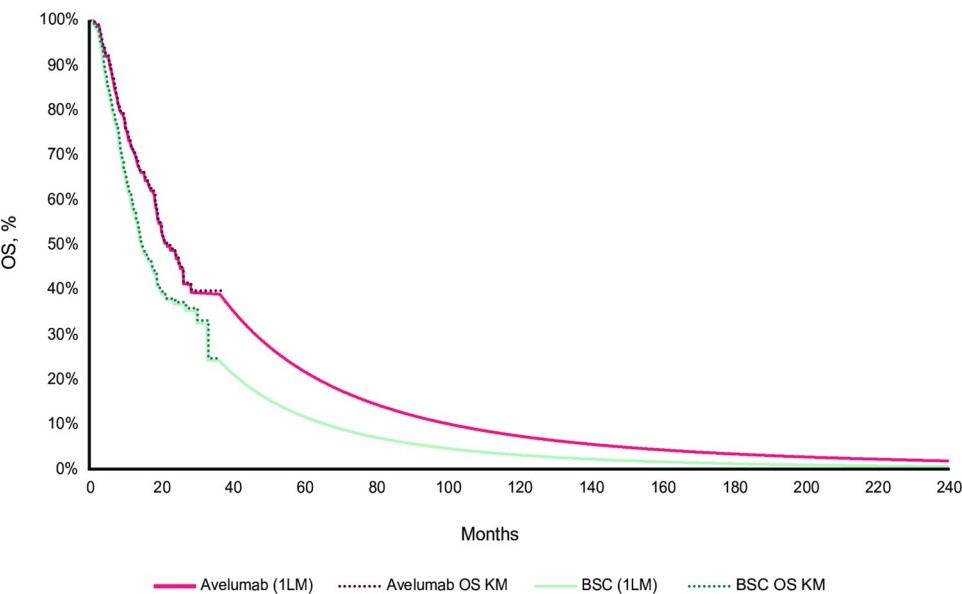

**Fig 4. OS KM curves followed by log-normal extrapolation beyond KM curves for both arms (reference analysis).**
1LM, first-line maintenance; BSC, best supportive care; KM, Kaplan-Meier; OS, overall survival.

PFS data for both arms were extrapolated independently because they did not meet the proportional hazards assumption. Beyond the KM curves, a Weibull distribution and a log-normal distribution were considered for the avelumab plus BSC and BSC alone arms, respectively. In the avelumab plus BSC arm, the 3 best models according to Akaike information criteria and Bayesian information criteria were the log-normal, generalized gamma, and log-logistic distributions. However, with these models, the PFS curve that crossed the avelumab plus BSC OS at the end of the time horizon. To ensure that the model was clinically plausible, the Weibull distribution was chosen in the avelumab plus BSC arm because it had a good visual fit to the observed KM curve data [9]. This also yielded more conservative results in terms of long-term survival than the other 3 distributions mentioned above. The TTD data were considered to account for treatment discontinuations. A generalized gamma distribution and a log-logistic distribution were considered for TTD for the avelumab plus BSC and BSC alone arms, respectively, because the proportional hazards assumption was not met. Post-progression treatment distribution and post-progression duration of treatment were estimated based on trial data. KM curves and extrapolations for OS, PFS, and TTD in the base case analysis are presented in Figs 4–6.

## Costs and utilities

As defined by the HAS methodological guidelines for the cost-effectiveness analysis, the economic analysis was performed from a perspective that included the direct costs incurred by all relevant stakeholders, referred to as the collective perspective [17]. Direct medical costs included treatment-related (costs of treatment acquisition, administration, and subsequent treatments), follow-up, AE management, transportation, and end-of-life costs (Table 2). The unit costs were inflated to 2019 using the published French consumer price index [20–22]. In this analysis, costs and outcomes were discounted at 2.5% per year, as per the current methodological guidelines in France [17].

**Acquisition, administration costs, and dispensing fees.** Acquisition costs were based on publicly available data in France and included all taxes [23]. The avelumab cost was based on

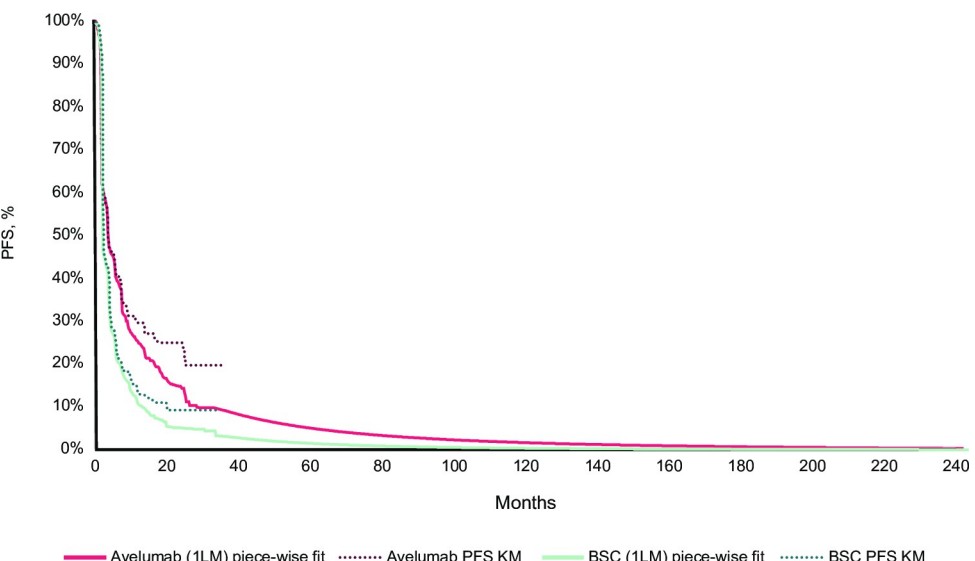

**Fig 5. PFS KM curves followed by Weibull and log-normal extrapolations for avelumab plus BSC and BSC alone, respectively (reference analysis).** 1LM, first-line maintenance; BSC, best supportive care; KM, Kaplan-Meier; PFS, progression-free survival.

the price negotiated with French authorities in September 2022 [24]. No administration costs were included for oral, nasal, ocular, or auricular treatments; however, a dispensing cost was included, in line with the packaging and dispensing fees set by French authorities [25]. Because no precise definition of BSC for UC was identified in published literature, the acquisition cost of BSC alone was estimated from the 10 categories of concomitant therapies most frequently reported in the JAVELIN Bladder 100 study, according to the Anatomical Therapeutic

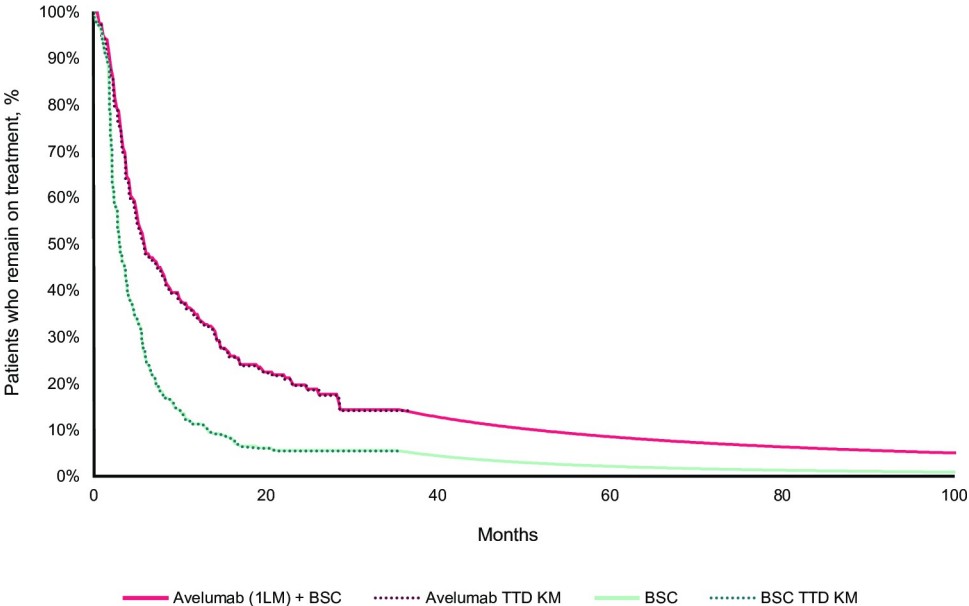

**Fig 6. TTD KM curves followed by generalized gamma and log-logistic extrapolations for avelumab plus BSC and BSC alone, respectively (reference analysis).** 1LM, first-line maintenance; BSC, best supportive care; KM, Kaplan-Meier; TTD, time to treatment discontinuation.

**Table 2. Summary of costs included in the model for each cost category and by treatment arm.**

| Treatment Arm | Avelumab plus BSC | BSC alone |
|---|---|---|
| **Acquisition cost** | €3,427.96 per administration (every two weeks) | €37.83 per administration (every month) |
| **Administration cost** | €413.07 per administration | €0.00 per administration |
| **Transportation cost** | €54.04 round trip | |
| **Adverse event costs** | • Lipase increase: €671.63<br>• Amylase increase: €671.63<br>• Anemia: €2,617.96<br>• Immune-related rash: €2,105.52<br>• Immune-related erythema multiforme: €2,105.52<br>• Immune-related maculopapular rash: €2,105.52<br>• Immune-related type 1 diabetes: €2,556.44<br>• Immune-related myositis: €2,345.57<br>• Nausea and vomiting: €1,291.46<br>• Injection site reaction: €3,564.41<br>• CPK increase: €671.63<br>• Neutropenia: €1,197.44<br>• Troponin T increase: €1,507.51<br>• Hyperglycemia: €2,556.44<br>• Hypophosphatemia: €2,926.07<br>• Arthralgia: €1,499.35<br>• Headache: €1,203.30<br>• Hypertension: €2,523.26<br>• Diarrhea (grade 1–2): €84.87<br>• Fatigue (grade 1–2): €93.65 | |
| **Follow-up costs** | Medical consultations<br>• General practitioner: €25.42<br>• Urological surgeon: €79.85<br>• Medical oncologist: €34.92<br>• Radiologist: €312.98<br>• Nephrologist: €145.34<br>• Nuclear medicine physician: €239.25<br>• Radiotherapist: €34.11<br>Biological examinations<br>Blood count: €6.75<br>Creatinine clearance: €1.62<br>• Complete ionogram: €5.94<br>• ALAT/ASAT: €2.70<br>• GGT: €1.62<br>• Platelets: €4.05<br>• Technical package: €9.72<br>• Technical procedures<br>• Scanner/tomodensitometry: €106.22<br>• Urethrocystoscopy: €144.00 | Medical consultations<br>• General practitioner: €25.42<br>• Urological surgeon: €79.85<br>• Medical oncologist: €34.92<br>• Radiologist: €312.98<br>• Nephrologist: €145.34<br>• Nuclear medicine physician: €239.25<br>• Radiotherapist: €34.11<br>Biological examinations<br>• Blood count: €6.75<br>• Creatinine clearance: €1.62<br>• Complete ionogram: €5.94<br>• ALAT/ASAT: €2.70<br>• GGT: €1.62<br>• Platelets: €4.05<br>• Technical package: €9.72<br>Technical procedures<br>• Scanner/tomodensitometry: €106.22 |
| **Post-progression treatment costs** | Immunotherapy: €5,525 over the entire post-progression duration of the avelumab plus BSC arm<br>(16.68 weeks)<br>• Pembrolizumab: €1,926.37 per cycle<br>Chemotherapy: €16,018 over the entire post-progression duration of the avelumab plus BSC arm<br>(15.99 weeks)<br>• Paclitaxel: €586.44 per cycle<br>• Gemcitabine: €879.66 per cycle<br>• Cisplatin: €293.22 per cycle<br>• Carboplatin: €879.66 per cycle<br>• Vinflunine: €390.96 per cycle | Immunotherapy: €35,275 over the entire post-progression duration of the BSC arm<br>(25.12 weeks)<br>• Pembrolizumab: €1,926.37 per cycle<br>Chemotherapy: €9,056 over the entire post-progression duration of the BSC arm<br>(16.60 weeks)<br>• Paclitaxel: €586.44 per cycle<br>• Gemcitabine: €879.66 per cycle<br>• Cisplatin: €293.22 per cycle<br>• Carboplatin: €879.66 per cycle<br>• Vinflunine: €390.96 per cycle |
| **End-of-life costs** | €6,695.53 per patient | |

ALAT, alanine aminotransferase; ASAT, aspartate aminotransferase; BSC, best supportive care; CPK, phosphocreatine kinase; GGT, gamma-glutamyl transferase.

Chemical (ATC) level 2 classification system. The dosage reported in the Summary of Product Characteristics for each medication was considered. The medication with the highest number of boxes sold in each category (ATC level 2 classification), which was reported in the Medic'AM 2019 database, was included in the BSC [26]. When the international non-proprietary name did not clearly identify a molecule, the brand name of the product with the highest number of boxes sold was used. The monthly cost of BSC alone was estimated at €37.83. For treatments administered intravenously, an injection procedure carried out in the day hospital was accounted for. The administration costs were estimated from the average cost of a stay for a chemotherapy session, weighted by the number of stays in public and private health facilities reported by ScanSanté [27, 28]. The analysis assumed no vial sharing between patients.

**Transportation costs.**  A transport cost was applied for the management of AEs requiring hospitalization and the administration of intravenous treatments. The average cost of a one-way service according to the mode of transport (ambulance, taxi, or light medical vehicle) was estimated based on the literature [28–32]. The proportion of patients benefiting from a reimbursed or nonreimbursed mode of transport was also considered [33].

**Adverse events costs.**  All Grade ≥3 treatment-related AEs were included in the model with minimum frequency. Grade 1–2 AEs that could impact costs or QoL were also included in the model. For each treatment arm, the incidence rate associated with the occurrence of ≥1 episode of the AE was estimated based on data from the JAVELIN Bladder 100 study. The model then calculated an incidence rate per cycle that was applied over the duration of treatment with avelumab plus BSC or BSC alone. It was assumed that any grade 3 or higher AEs systematically led to a hospitalization according to the definition of the CTCAE of the National Cancer Institute [34]. Hospitalization costs were derived from the latest available French National Cost Study (Étude Nationale des Coûts), which provides cost data per Groupe Homogène de Malades (similar to diagnosis-related groups) from a representative sample of public and private hospitals [35], with costs based on the literature [36].

**Post-progression treatments costs.**  The cost of post-progression treatments, i.e., in second and subsequent lines (2L+) of treatment, were included as a basket of treatments received after disease progression in the JAVELIN Bladder 100 trial and were applied per cycle. Trial patients could receive immunotherapy with atezolizumab, pembrolizumab, durvalumab, or nivolumab as second-line (2L) treatment. However, in France, only pembrolizumab is reimbursed as a 2L treatment for mUC in the framework of a derogation process [37]. Therefore, patients receiving atezolizumab, durvalumab, or nivolumab were reassigned to pembrolizumab to reflect French practice (Table 3). The average cost of the treatment basket was calculated as the product of the weekly treatment cost and the duration of post-progression treatment, assuming that the efficacy of 2L+ treatments was represented by the efficacy of JAVELIN Bladder 100 post-progression treatments. Because patients could receive several treatments at the same time, the distribution of treatments received post-progression could exceed 100%.

**Follow-up costs.**  Medical consultations, biological tests, and medical and imaging procedures were included in the model as follow-up costs. The nature and frequency of healthcare resources used were determined based on the HAS recommendations for bladder cancer and the French National Cancer Institute recommendations [38, 39]. To support the information identified in the literature, a questionnaire was completed by the clinical experts contacted.

**End-of-life costs.**  End-of-life costs were included in the model as a one-off cost. The average end-of-life cost according to the place of care (hospital, home, nursing home, etc.) was weighted by the proportion of patients who died in each place of care [28, 40].

**Utilities.**  Health-related QoL data were collected from the JAVELIN Bladder 100 trial using the EQ-5D-5L questionnaire at inclusion, on the first day of each treatment cycle (4

**Table 3. Distribution of post-progression treatments in JAVELIN Bladder 100 compared with the cost-effectiveness model.**

| Post-Progression Treatment (2L+) | Maintenance Treatment Received and Included in the Cost-Effectiveness Model | |
|---|---|---|
| | Avelumab plus BSC | BSC alone |
| **Immunotherapy, %** | | |
| Atezolizumab | 0 | 0 |
| Nivolumab | 0 | 0 |
| Pembrolizumab | 17.2 | 72.9 |
| Durvalumab | 0 | 0 |
| **Chemotherapy, %** | | |
| Cisplatin | 20.0 | 9.9 |
| Carboplatin | 31.7 | 17.1 |
| Gemcitabine | 42.1 | 25.1 |
| Paclitaxel | 33.1 | 19.4 |
| Vinflunine | 25.5 | 8.1 |
| **Total, %** | **170** | **153** |

2L+, second and subsequent lines; BSC, best supportive care.

weeks), at the end of treatment and/or withdrawal of consent, and at 30-, 60-, and 90-day follow-up visits. The utility values incorporated in the model were valued based on French tariffs with a direct approach using the value from a representative sample of the French population set derived by Andrade et al. [41], and using linear mixed-effects models, including baseline utility and progression status as covariates (Table 4). The JAVELIN Bladder 100 study design did not accurately capture disutilities associated with AEs. Therefore, a targeted literature review (TLR) was conducted for this purpose. The disutilities from the Beusterien et al. [42], Nafees et al. [43], and Lloyd et al. [44] studies were considered for diarrhea, anemia, fatigue, and rash (Table 5). The mean of the disutilities associated with the AEs identified in the literature was considered for the AEs for which no disutilities could be identified; the mean was estimated to 0.083. The costs and disutilities were applied for each cycle according to the rate of occurrence of the AE in each treatment arm (grade 1–2 and grade $\geq$3 AEs).

## Base case analysis and sensitivity analyses

The results of the economic analysis were expressed as costs per quality-adjusted life-year (QALY) gained and costs per life-year (LY) gained. A one-way deterministic sensitivity analysis was conducted to evaluate the influence of each individual parameter uncertainty on model outcomes. A range of ±10% of the base value was explored for parameters without information on the standard error or CI. A probabilistic sensitivity analysis was performed to evaluate the

**Table 4. Summary of utility data for avelumab plus BSC and BSC alone considered for the reference analysis.**

| Health State | Utility Value (95% CI) |
|---|---|
| **Overall population** | |
| PFS[a] | 0.894 (0.883–0.905) |
| PPS[a] | 0.840 (0.828–0.852) |

BSC, best supportive care; PFS, progression-free survival; PPS, post-progression survival.

[a] Data obtained from the JAVELIN Bladder 100 trial.

**Table 5. Disutilities associated with adverse events of treatments studied for efficiency analysis in the reference analysis.**

| Adverse Event | Disutility[a] | SE | Source |
|---|---|---|---|
| Anemia | 0.115 | NR | Lloyd A, et al. Health state utilities for metastatic breast cancer. Br J Cancer. 2006;95(6):683–90. |
| Immune-related rash | 0.032 | 0.012 | Nafees B, et al. Health state utilities for non small cell lung cancer. Health Qual Life Outcomes. 2008;6(1):84. |
| Diarrhea | 0.103 | NR | Lloyd A, et al. Health state utilities for metastatic breast cancer. Br J Cancer. 2006;95(6):683–90. |
| Fatigue | 0.115 | NR | |
| Grade 1–2 diarrhea | 0.080 | 0.020 | Beusterien KM, et al. Population preference values for treatment outcomes in chronic lymphocytic leukaemia: a cross-sectional utility study. Health Qual Life Outcomes. 2010;8(1):50. |
| Grade 1–2 fatigue | 0.115 | NR | Lloyd A, et al. Health state utilities for metastatic breast cancer. Br J Cancer. 2006;95(6): 683–90. |

AE, adverse event; NR, not reported; SE, standard error.

[a] For AEs for which no disutility could be identified in the literature, the average of the identified AE disutilities was considered.

overall impact of parameter uncertainty in the model. One thousand Monte Carlo simulations were run, sampling from the defined probability distributions each time. Gamma distributions were assumed for costs, and beta distributions were assumed for utility values and probabilities. The results were presented as a cost-effectiveness acceptability curve. A total of 51 scenario analyses were also conducted to assess the uncertainty associated with the choice of extrapolations in the base case analysis, the approach used to estimate utilities in the model, the distribution of post-progression treatments, and the variation of the time horizon.

**Model validation.** An internal model validation was performed by a scientific committee, which included external experts in health economics, health technology assessments, and statistics, who assessed the model structure, model inputs, and the apparent validity. An external validation of clinical data and real-world evidence data was conducted using two studies [45, 46]. The Leeds Cancer Center cohort study in the United Kingdom reported a 5-year OS rate of 20% in patients with mUC treated with cisplatin or carboplatin plus gemcitabine [46]. A European Organization for Research and Treatment of Cancer study in patients with mUC treated with cisplatin plus gemcitabine and carboplatin plus gemcitabine reported a 5-year OS rate of 11% and a 5-year PFS rate of 4% [45, 47]. Although the populations in these 2 studies were not fully comparable to those in JAVELIN Bladder 100, these studies were chosen for external validation because they were the only studies with long follow-up at the time of the HAS submission.

## Results

### Base case analysis

Results for the reference case are presented in Table 6. Patients in the avelumab plus BSC arm lived for an average of 2.91 years versus 2.14 years in the BSC alone arm, yielding an increase of 0.77 discounted LYs (+36.2%) favouring avelumab 1L maintenance plus BSC strategy,

**Table 6. Discounted results of the reference analysis (ICER).**

| | Total cost, € | QALYs | LYG | Incremental | | | ICER | |
|---|---|---|---|---|---|---|---|---|
| | | | | Cost, € | QALYs | LYG | €/LYG | €/QALY |
| **Avelumab plus BSC** | 136,917 | 2.4918 | 2.9126 | 97,166 | 0.6672 | 0.7747 | 125,429 | 145,626 |
| **BSC alone** | 39,751 | 1.8246 | 2.1379 | | | | | |

BSC, best supportive care; ICER, incremental cost-effectiveness ratio; LYG, life-years gained; QALY, quality-adjusted life-year.

which was directly related to the OS benefits of avelumab. After adjustment for QoL, avelumab plus BSC and BSC alone resulted in 2.49 and 1.82 QALYs, respectively, yielding an increase of 0.67 discounted QALYs in favor of avelumab plus BSC (+37%).

Over a 10-year life horizon, the discounted lifetime costs of avelumab 1L maintenance therapy plus BSC and BSC alone were €136,917 and €39,751, respectively. The difference in cost between the two strategies was €97,166 and was primarily due to the cost of acquisition and administration of avelumab. However, some offsets of –€20,424 and –€351 were estimated for post-progression and end-of-life costs, respectively. The projected higher lifetime cost of avelumab plus BSC, in conjunction with its projected incremental gain in QALYs, yields an incremental cost-effectiveness ratio (ICER) of €145,626 per QALY. The corresponding ICER measured in € per LY gained was slightly lower (€125,429).

## Sensitivity analyses

A scenario analysis considered that a higher proportion of patients received PD-1/PD-L1 immunotherapy post progression in the BSC alone arm (80%) compared with the avelumab plus BSC arm (10%). The scientific committee stated that it was unlikely that >80% of patients would receive PD-1/PD-L1 immunotherapy following BSC alone and that fewer patients who discontinued avelumab due to severe toxicity would receive a second immunotherapy after avelumab in France. This assumption showed a slight variation in the ICER (€141,726/QALY or −2.7%) compared with that in the base case analysis. Various scenarios were considered that yielded an ICER variation of >20%: a TTD exponential distribution for avelumab plus BSC (−24.9%), a 20% decrease in pre- and post-progression utility values (+25.5%), a cross-walk algorithm approach for utilities (+22.4%) [48], a 5-year time horizon (+22.4%), and an OS Weibull distribution for both arms (+29.1%).

The one-way sensitivity analysis results are presented in a tornado diagram (Fig 7). The results demonstrated that OS had the greatest influence on the ICER, varying between €122,927 and €180,194, corresponding to a change of −15.6% and +23.7%, respectively. Other parameters that had the greatest impact on the ICER were time receiving post-progression immunotherapy in the BSC alone arm, cost of post-progression treatments per cycle for BSC alone, and avelumab plus BSC efficacy parameters for OS and TTD.

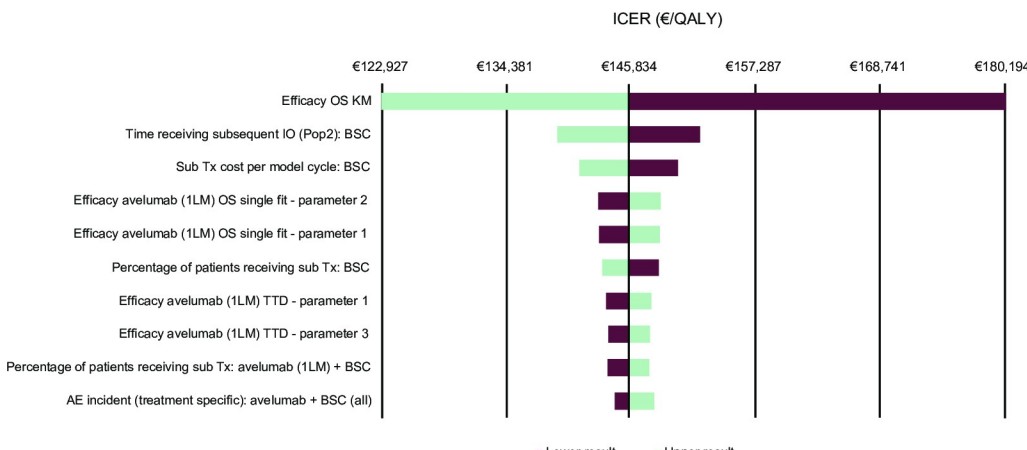

**Fig 7. Tornado diagram for the one-way sensitivity analysis.** 1LM, first-line maintenance, AE, adverse event; BSC, best supportive care; ICER, incremental cost-effectiveness ratio; IO, immuno-oncotherapy; KM, Kaplan-Meier; OS, overall survival; pop2, population 2; QALY, quality-adjusted life-year; sub, subsequent; TTD, time to treatment discontinuation; Tx, treatment.

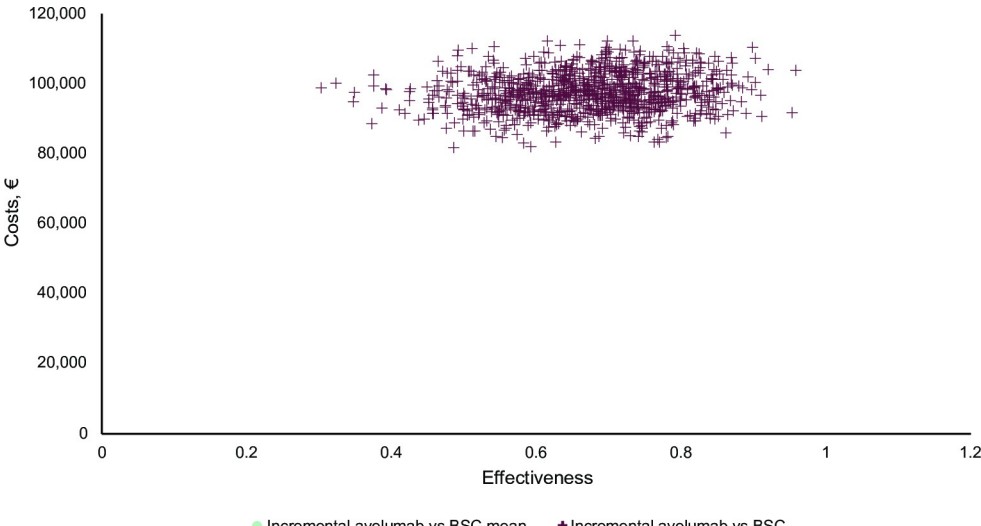

**Fig 8. Scatterplot of the probabilistic sensitivity analysis comparing avelumab plus BSC versus BSC alone.** BSC, best supportive care.

The results of the probabilistic sensitivity analysis show that the average incremental cost of 1,000 Monte Carlo simulations was €97,344 with an average of 0.6731 incremental QALYs gained, corresponding to an ICER of €144,631/QALY gained; this is in line with the deterministic base case analysis (−0.7%). All simulations were in the northeast quadrant of the cost-effectiveness plane, indicating that avelumab plus BSC is more effective and expensive than BSC alone (Fig 8). The cost-effectiveness acceptability curve demonstrated that avelumab 1L maintenance therapy plus BSC was associated with an 81% probability of being cost-effective versus BSC alone at the current willingness-to-pay threshold of €300,000/QALY (Fig 9).

## Discussion

This study demonstrated that avelumab 1L maintenance therapy plus BSC, as compared with BSC alone, in eligible adults with la/mUC in France was associated with an ICER of €145,626/

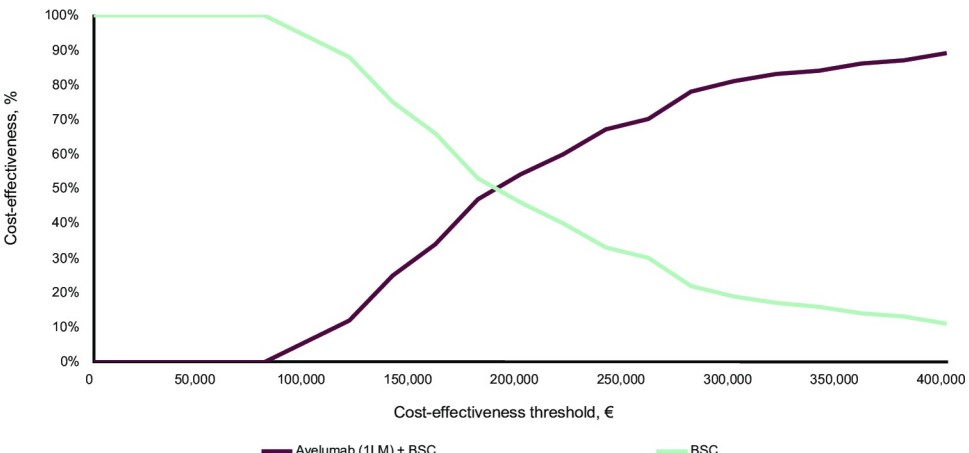

**Fig 9. Acceptability curve of cost-effectiveness for the reference analysis.** 1LM, first-line maintenance; BSC, best supportive care.

QALY gained. The ICER value falls within the lower end of the recently published range for the value of a QALY in France of €147,093 to €201,398 [49]. These findings are also consistent with ICERs for other country models for avelumab 1L maintenance therapy plus BSC in patients with mUC in the US [50] and China [51], which ranged between $102,365/QALY and $241,610/QALY. Similar incremental QALYs and LYs were also reported for the United Kingdom, Finland, and Taiwan [52–54]. Avelumab 1L maintenance therapy plus BSC compared with BSC alone was associated with improved life expectancy with incremental QALYs and LYs gained of 0.67 and 0.77, respectively. ICER variations from the base case analysis ranged from −24.9% to +29.1%. To determine TTD variations, the exponential distribution in the avelumab plus BSC arm was used; to determine the OS variation, the Weibull distribution was used for both arms. All parameters used in the analyses were based on the HAS 2020 methodological guidelines [17]. This economic analysis used clinical data derived from the JAVELIN Bladder 100 trial and published literature [8]. The trial included a high number of patients enrolled in France (n = 82) and their characteristics were consistent with those of more than 200 French patients from a retrospective chart review conducted as part of the present analyses [14].

Our study was based on a direct comparison of avelumab plus BSC with BSC alone; BSC alone was the standard treatment in recommendations until the results for avelumab plus BSC were published. The JAVELIN Bladder 100 study demonstrated the superiority of avelumab plus BSC versus BSC alone in terms of OS. The extrapolations of the OS data were also validated by a scientific committee, which was composed of three clinical experts and an economic expert. All scenario analyses related to the choice of OS distribution beyond the KM curves fell within an acceptable ICER threshold, although no official ICER threshold has been defined in France. In addition, two studies from the Leeds Cancer Center and the European Organisation for Research and Treatment of Cancer provided external validation [45–47]. The extrapolations of the selected PFS and TTD data represented conservative choices. Based on the scenario analyses performed, the distribution selected in the base case analysis to model PFS beyond KM curves was the most conservative. The scenario analyses performed to test different TTD extrapolations showed negligible variation in ICERs in the BSC alone arm. The TTD distribution chosen for the avelumab plus BSC arm was the second-most conservative.

The utility data from the EQ-5D-5L questionnaire administered to patients in the JAVELIN Bladder 100 study were weighted according to French preferences via the direct approach [40], as recommended by the HAS 2020 methodological guidelines. All grade ≥3 AEs from JAVELIN Bladder 100 were accounted for in terms of costs and QoL and grade 1–2 AEs with the highest impact on QoL were also accounted for. Finally, the different scenario analyses conducted demonstrated the robustness of the results, with an increase or a decrease of 5% of the ICER for most scenarios.

This analysis had some limitations. The parametric functions of the survival curves corresponding to best statistical fit resulted in the crossing of the PFS and OS curves beyond the KM curve and were not clinically possible. As a result, they were considered in the scenario analyses with adjustment. The Weibull distribution was retained in the reference analysis to model the PFS of avelumab plus BSC beyond the KM curve. To compensate for this uncertainty, a time horizon of 10 years, shorter than lifetime, was considered in the reference analysis, which limited the uncertainty associated with extrapolations. The choice of extrapolations was also validated by experts from the scientific committee. The scenario analyses performed with the second, third, fourth, and fifth best-fitting distributions showed little variation in the ICER for PFS in the avelumab plus BSC arm. In addition, several assumptions were made about the treatments considered in the BSC alone arm due to the uncertainty and the important heterogeneity related to the composition of BSC alone between patients. Nevertheless, a sensitivity analysis with a 100% change in cost was performed, which showed a slight

difference in the ICER (±0.3%). AE-related disutilities were not specific to UC; the TLR did not identify any in the indication. Therefore, two sensitivity analyses were performed, one with varying disutility values and another not considering disutilities. The scenario analysis that did not consider the AE-related disutilities did not show a major impact on the ICER (−1.4%).

Because avelumab is the first 1L maintenance therapy for la/mUC, few external real-world data were available to validate the long-term extrapolations at the time of the HAS submission in 2021, and a TLR did not identify any relevant data. The external validation has certain limitations; specifically, it was performed using publications of patients who had received 1L treatment (not 1L maintenance therapy), and no cross-validation could be performed because no other cost-effectiveness analysis was carried out in this indication. However, the extrapolations, resources consumed, and post-progression treatments were validated by three clinical experts in the scientific committee. Two sensitivity analyses that varied the cost associated with the resources consumed and the post-progression treatments were performed. Although the cost of post-progression treatments in the BSC alone arm appeared to be one of the factors influencing the ICER, the variation remained small (−2.7%). In addition, AVENANCE (NCT04822350), an ongoing, noninterventional, real-world, ambispective study, reported preliminary results of the real-world use of avelumab as a 1L maintenance therapy in patients with la/mUC in France. A preliminary analysis reported a 12-month OS rate of 64.1% (95% CI, 57.1%-70.3%) and a median PFS from the start of avelumab of 5.7 months (95% CI, 5.1–7.9) [55]. These findings are consistent with JAVELIN Bladder 100 results and support the real-world effectiveness of avelumab 1L maintenance in patients with la/mUC without disease progression after 1L platinum-based chemotherapy.

Regarding contextual limitations, the estimation of uncertain parameters was based on a choice justified by either their transposability to current practice or by their methodological robustness. When such references were not available, conservative choices were made, including, but not limited to, the choice of a joint log-normal model to estimate OS and the method of estimating the cost of acquiring BSC.

## Conclusion

Avelumab 1L maintenance plus BSC was associated with an ICER of €145,626/QALY with a total incremental cost of €97,166 per patient over a 10-year time horizon. Avelumab plus BSC was associated with health benefits in terms of QALYs and LYs of 0.67 and 0.77, respectively. The analysis robustly estimated the cost-utility ratio of avelumab plus BSC versus BSC alone in patients with la/mUC that had not progressed following platinum-based chemotherapy, and showed low uncertainty associated with the endpoints and outcomes (±5%). The sensitivity and scenario analyses performed in the context of this economic evaluation provided elements that enabled the level of uncertainty to be comprehensively documented. The efficiency of avelumab 1L maintenance plus BSC in eligible patients with la/mUC was also overall positively evaluated by the HAS with few reservations [56].

## Acknowledgments

Medical writing was provided by Paola Marino and Sudipta Ridhurkar from Amaris Consulting and editorial support was provided by Nucleus Global.

## Author Contributions

**Conceptualization:** Fanny Porte, Anna Granghaud, Jane Chang, Mairead Kearney, Aya Morel, Ingrid Plessala, Hélène Cawston, Julie Roiz, Ying Xiao, Prisca Lambert, Alain Ravaud, Yohann Loriot, Pierre Lévy.

**Data curation:** Jane Chang, Antoine Thiery-Vuillemin.

**Formal analysis:** Mairead Kearney, Ingrid Plessala, Hélène Cawston, Julie Roiz, Ying Xiao, Alain Ravaud, Pierre Lévy.

**Investigation:** Ingrid Plessala, Hélène Cawston, Antoine Thiery-Vuillemin.

**Methodology:** Fanny Porte, Anna Granghaud, Jane Chang, Mairead Kearney, Ingrid Plessala, Hélène Cawston, Julie Roiz, Ying Xiao, Alain Ravaud, Pierre Lévy.

**Project administration:** Ingrid Plessala, Hélène Cawston.

**Resources:** Aya Morel.

**Supervision:** Fanny Porte, Anna Granghaud, Jane Chang, Ingrid Plessala, Hélène Cawston, Julie Roiz, Ying Xiao.

**Validation:** Fanny Porte, Anna Granghaud, Aya Morel, Ingrid Plessala, Hélène Cawston, Julie Roiz, Ying Xiao, Marie-Noelle Solbes, Prisca Lambert, Yohann Loriot, Antoine Thiery-Vuillemin, Pierre Lévy.

**Visualization:** Antoine Thiery-Vuillemin.

**Writing – original draft:** Ingrid Plessala, Hélène Cawston.

**Writing – review & editing:** Fanny Porte, Anna Granghaud, Jane Chang, Mairead Kearney, Ingrid Plessala, Hélène Cawston, Julie Roiz, Ying Xiao, Marie-Noelle Solbes, Prisca Lambert, Alain Ravaud, Yohann Loriot, Antoine Thiery-Vuillemin, Pierre Lévy.

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
