## [Decision Letter · Decision Letter 0]

15 Aug 2023

PONE-D-23-16035Cost-effectiveness of avelumab first-line maintenance therapy for adult patients with locally advanced or metastatic urothelial carcinoma in FrancePLOS ONE

Dear Dr. Plessala,

Thank you for submitting your manuscript to PLOS ONE. After careful consideration, we feel that it has merit but does not fully meet PLOS ONE’s publication criteria as it currently stands. Therefore, we invite you to submit a revised version of the manuscript that addresses the points raised during the review process. This work is of high quality. However, I have a major concern about the availability of data. It should be included as a supplementary file or in a public repository. This work was conducted by researchers working for the pharmaceutical industry and therefore there is a potential conflict of interest. According to the explanation in the data sharing statement, the data will be shared only in specific cases. In terms of transparency, the data used for the analysis should be freely available.

Please submit your revised manuscript by Sep 29 2023 11:59PM. If you will need more time than this to complete your revisions, please reply to this message or contact the journal office at plosone@plos.org. Please include the following items when submitting your revised manuscript:A rebuttal letter that responds to each point raised by the academic editor and reviewer(s). You should upload this letter as a separate file labeled 'Response to Reviewers'.A marked-up copy of your manuscript that highlights changes made to the original version. You should upload this as a separate file labeled 'Revised Manuscript with Track Changes'.An unmarked version of your revised paper without tracked changes. You should upload this as a separate file labeled 'Manuscript'.If applicable, we recommend that you deposit your laboratory protocols in protocols.io to enhance the reproducibility of your results. Protocols.io assigns your protocol its own identifier (DOI) so that it can be cited independently in the future. For instructions see: https://journals.plos.org/plosone/s/submission-guidelines#loc-laboratory-protocols. Additionally, PLOS ONE offers an option for publishing peer-reviewed Lab Protocol articles, which describe protocols hosted on protocols.io. Read more information on sharing protocols at https://plos.org/protocols?utm_medium=editorial-email&utm_source=authorletters&utm_campaign=protocols.

We look forward to receiving your revised manuscript.

Kind regards,

Joseph Pinto

Academic Editor

PLOS ONE

“This study was funded by Merck Santé S.A.S, an affiliate of Merck KGaA, Darmstadt, Germany (CrossRef Funder ID: 10.13039/100009945), as part of an alliance between the healthcare business of Merck KGaA, Darmstadt, Germany and Pfizer.”

“This study was funded by Merck Santé S.A.S, an affiliate of Merck KGaA, Darmstadt, Germany (CrossRef Funder ID: 10.13039/100009945), as part of an alliance between the healthcare business of Merck KGaA, Darmstadt, Germany and Pfizer. Medical writing was provided by Paola Marino and Sudipta Ridhurkar from Amaris Consulting and editorial support was provided by Clinical Thinking, funded by Merck Santé S.A.S, an affiliate of Merck KGaA, Darmstadt, Germany, and Pfizer.”

“This study was funded by Merck Santé S.A.S, an affiliate of Merck KGaA, Darmstadt, Germany (CrossRef Funder ID: 10.13039/100009945), as part of an alliance between the healthcare business of Merck KGaA, Darmstadt, Germany and Pfizer.Please include your amended statements within your cover letter; we will change the online submission form on your behalf.”

“F. Porte is an employee of Merck Santé S.A.S., Lyon, France, an affiliate of Merck KGaA, Darmstadt, Germany at the time of the project. A. Granghaud was an employee of Pfizer S.A.S., Paris, France at the time of the study. J. Chang is an employee of Pfizer and holds stock and other ownership interest with Bayer, Bristol Myers Squibb, and Pfizer.  M. Kearney is an employee of Merck KGaA, Darmstadt, Germany, and holds stock in Merck KGaA, Darmstadt, Germany, Novartis and UCB. A. Morel is an employee of Pfizer S.A.S., Paris, France.

I. Plessala was an employee of Amaris Consulting, Paris, France at the time of the study.

H. Cawston is an employee of Amaris Consulting, Paris, France. J. Roiz is an employee of and reposts stocks and other ownership interest with Evidera. Y. Xiao is an employee of Evidera. M.-N. Solbes is an employee of Merck Santé S.A.S., Lyon, France, an affiliate of Merck KGaA, Darmstadt, Germany. P. Lambert is an employee of Pfizer S.A.S., Paris, France. A. Ravaud has received grants or contracts from Merck KGaA, Darmstadt, Germany, and Pfizer; has received travel and accommodation expenses from Ipsen Merck KGaA, Darmstadt, Germany, Merck & Co., Kenilworth, NJ, and Pfizer; and has participated in advisory boards for Esai, Ipsen, Merck KGaA, Darmstadt, Germany, and Pfizer. Y. Loriot has served in consulting or advisory roles for Astellas Pharma, Bristol Myers Squibb, Immunomedics, Janssen, Loxo/Lilly, Merck KGaA, Darmstadt, Germany, Merck & Co., Kenilworth, NJ, Pfizer, Roche, Seattle Genetics, and Taiho Pharmaceutical; has received travel and accommodations expenses from Astellas Pharma, Janssen Oncology,  Merck & Co., Kenilworth, NJ, Roche, and Seattle Genetics; and has received institutional research funding from Astellas Pharma, Basilea, Bristol Myers Squibb, Exelixis, Gilead Sciences, Incyte, Janssen Oncology, Merck KGaA, Darmstadt, Germany, Merck & Co., Kenilworth, NJ, Nektar, Pfizer, Roche, Sanofi, Seattle Genetics, and Taiho Pharmaceutical. A. Thiery-Vuillemin has participated in advisory boards for Astellas Pharma, AstraZeneca, Bristol-Myers Squibb, Ipsen, Janssen, Merck & Co., Kenilworth, NJ, Novartis, Pfizer, Roche/Genentech and Sanofi; reports employment by Bristol Myers Squibb; has served on steering committees for AstraZeneca, Bristol-Myers Squibb and Novartis; has received institutional research funding from Bayer, Ipsen and Pfizer; has served as principal investigator for Astellas Pharma, AstraZeneca, Bristol-Myers Squibb, Excelixis, Incyte, Ipsen, Johnson & Johnson, Merck & Co., Kenilworth, NJ, Novartis, Pfizer, Roche, Sanofi, and UNICANCER/GETUG; has received travel and accommodation expenses from Astellas Pharma, AstraZeneca, Bristol-Myers Squibb, Ipsen, Johnson & Johnson, Merck & Co., Kenilworth, NJ, Pfizer and Roche; and is a member of ASCO and GETUG. P. Lévy has served in consulting or advisory role and had received honoraria from Merck KGaA, Darmstadt, Germany.”

Additional Editor Comments:

This work is of high quality. However, I have a major concern about the availability of data. It should be included as a supplementary file. This work was conducted by researchers working for the pharmaceutical industry and therefore there is a potential conflict of interest. According to the explanation in the data sharing statement, the data will be shared only in specific cases. In terms of transparency, the data used for the analysis should be freely available.

Reviewers' comments:

Reviewer's Responses to Questions

**Comments to the Author**

1. Is the manuscript technically sound, and do the data support the conclusions?

Reviewer #1: Yes

Reviewer #2: Yes

2. Has the statistical analysis been performed appropriately and rigorously? 

Reviewer #1: Yes

Reviewer #2: I Don't Know

3. Have the authors made all data underlying the findings in their manuscript fully available?

Reviewer #1: Yes

Reviewer #2: Yes

4. Is the manuscript presented in an intelligible fashion and written in standard English?

Reviewer #1: Yes

Reviewer #2: Yes

5. Review Comments to the Author

Reviewer #1: The article is a great piece of knowledge. Well written. A remarkable introduction and background, with well described methods and robust results. The article presented accurate graphs and conclussions are close related to the findings.

Reviewer #2: Dear Colleagues, thank you for the opportunity to review this interesting and well-written paper aiming to evaluate the cost-effectiveness of avelumab maintenance therapy plus BSC vs BSC alone for adults with locally advanced or metastatic urothelial carcinoma that had not progressed following platinum-based chemotherapy in France.

This type of analysis holds critical importance in informing decision makers and other stakeholders about the budgetary impact (not only cost) of innovative drugs, as well as serving as a tool to negotiate decreasing cost of expenditure when the results are not favorable.

The paper effectively and comprehensively outlines all the steps taken during the analysis. Although numerous adaptations are required to suit specific health systems, thresholds, costs, and legislations in other regions, it nonetheless offers a detailed guide that can be applied in various settings.

Having stated the above, I am afraid the extensive involvement of industry representatives can somehow undermine the credibility of the analysis. The participation of the scientific community in the validation of the model needs to be highlighted and explained in depth.

6. PLOS authors have the option to publish the peer review history of their article (what does this mean?). If published, this will include your full peer review and any attached files.

Reviewer #1: No

Reviewer #2: No

---

## [Author Response · Author response to Decision Letter 0]

28 Sep 2023

Please see uploaded file "Response to Reviewers_GEVD France UC CE Manuscript_27Sept23"

---

## [Editor Report · Decision Letter 1]

5 Mar 2024

PONE-D-23-16035R1Cost-effectiveness of avelumab first-line maintenance therapy for adult patients with locally advanced or metastatic urothelial carcinoma in FrancePLOS ONE

Dear Dr. Plessala,

Thank you for submitting your manuscript to PLOS ONE. After careful consideration, we feel that it has merit but does not fully meet PLOS ONE’s publication criteria as it currently stands. Therefore, we invite you to submit a revised version of the manuscript that addresses the points raised during the review process.

We look forward to receiving your revised manuscript.

Kind regards,

Yudai Ishiyama

Academic Editor

PLOS ONE

Journal Requirements:

Additional Editor Comments:

I guess all the comments given for the original manuscript is answered except for one.

In the comments for original submission, a reviewer has recommended that the raw data supporting the results of this study should be fully open to public. It is not shown whether the authors have any intention to do so in the R1 response.

If the authors choose not to make the database fully open, I have to consult another expert if this is appropriate.

---

## [Author Response · Author response to Decision Letter 1]

2 Apr 2024

Please see the uploaded file named "Response to Reviewers_France UC CE Manuscript_02APR24"

---

## [Editor Report · Decision Letter 2]

9 Apr 2024

Cost-effectiveness of avelumab first-line maintenance therapy for adult patients with locally advanced or metastatic urothelial carcinoma in France

PONE-D-23-16035R2

Dear Dr. Plessala,

We’re pleased to inform you that your manuscript has been judged scientifically suitable for publication and will be formally accepted for publication once it meets all outstanding technical requirements.

Kind regards,

Yudai Ishiyama

Academic Editor

PLOS ONE

Additional Editor Comments (optional):

I have no additional comments.